# NCBI’s Virus Discovery Codeathon: Building “FIVE” —The Federated Index of Viral Experiments API Index

**DOI:** 10.3390/v12121424

**Published:** 2020-12-10

**Authors:** Joan Martí-Carreras, Alejandro Rafael Gener, Sierra D. Miller, Anderson F. Brito, Christiam E. Camacho, Ryan Connor, Ward Deboutte, Cody Glickman, David M. Kristensen, Wynn K. Meyer, Sejal Modha, Alexis L. Norris, Surya Saha, Anna K. Belford, Evan Biederstedt, James Rodney Brister, Jan P. Buchmann, Nicholas P. Cooley, Robert A. Edwards, Kiran Javkar, Michael Muchow, Harihara Subrahmaniam Muralidharan, Charles Pepe-Ranney, Nidhi Shah, Migun Shakya, Michael J. Tisza, Benjamin J. Tully, Bert Vanmechelen, Valerie C. Virta, JL Weissman, Vadim Zalunin, Alexandre Efremov, Ben Busby

**Affiliations:** 1Laboratory of Clinical and Epidemiological Virology, KU Leuven Department of Microbiology, Immunology and Transplantation, Rega Institute, BE3000 Leuven, Belgium; ward.deboutte@kuleuven.be (W.D.); cody.glickman@cuanshutz.edu (C.G.); bert.vanmechelen@kuleuven.be (B.V.); 2Integrative Molecular and Biomedical Sciences Program, Baylor College of Medicine, Houston, TX 77030, USA; 3Margaret M. and Albert B. Alkek Department of Medicine, Nephrology, Baylor College of Medicine, Houston, TX 77030, USA; 4Department of Genetics, MD Anderson Cancer Center, Houston, TX 77030, USA; 5School of Medicine, Universidad Central del Caribe, Bayamón, PR 00960, USA; 6Genetics & Molecular Biology, Millersville University, 40 Dilworth Rd, Millersville, PA 17551, USA; sierradesireemiller@gmail.com; 7Department of Epidemiology of Microbial Diseases, Yale School of Public Health (YSPH), 60 College Street, New Haven, CT 06510, USA; anderson.brito@yale.edu; 8National Center for Biotechnology Information, U.S. National Library of Medicine, National Institutes of Health, 9000 Rockville Pike, Bethesda, MD 20894, USA; camacho@ncbi.nlm.nih.gov (C.E.C.); jamesbr@ncbi.nlm.nih.gov (J.R.B.); zaluninvv@ncbi.nlm.nih.gov (V.Z.); alexandre.efremov@nih.gov (A.E.); 9Computational Bioscience Program, University of Colorado Anschutz, Aurora, CO 80045, USA; dk131363@gmail.com; 10AAAS Science and Technology Policy Fellow, Office of Data Science Strategy, Division of Program Coordination, Planning, and Strategic Initiatives, Office of the Director, National Institutes of Health, 31 Center Dr., Bethesda, MD 20894, USA; wynn@lehigh.edu; 11MRC-University of Glasgow Centre for Virus Research, Glasgow G61 1QH, UK; s.modha.1@research.gla.ac.uk; 12Biotechnology Graduate Program, University of Maryland Global Campus, 1616 McCormick Drive, Largo, MD 20774, USA; alexisleighnorris@gmail.com; 13Boyce Thompson Institute, Ithaca, NY 14850, USA; ss2489@cornell.edu; 14School of Animal and Comparative Biomedical Sciences, The University of Arizona, Tucson, AZ 85721, USA; 15Laboratory of Cellular Oncology, National Cancer Institute, 37 Convent Dr., Bethesda, MD 20894, USA; belfordak@nih.gov (A.K.B.); mike.tisza@nih.gov (M.J.T.); 16Memorial Sloan Kettering Cancer Center, New York, NY 10065, USA; evan.biederstedt@gmail.com; 17School of Life and Environmental Sciences and School of Medical Sciences, Marie Bashir Institute for Infectious Diseases and Biosecurity, The University of Sydney, Sydney, Australia; jan.buchmann@sydney.edu.au; 18Department of Biomedical Informatics, University of Pittsburgh, Pittsburgh, PA 15260, USA; npc19@pitt.edu; 19College of Science and Engineering, Flinders University, Bedford Park, SA 5042, Australia; robert.edwards@flinders.edu.au; 20Department of Computer Science, University of Maryland, College Park, MD 20740, USA; kjavkar@cs.umd.edu (K.J.); hsmurali@cs.umd.edu (H.S.M.); nidhi@cs.umd.edu (N.S.); 21Joint Institute for Food Safety and Applied Nutrition, University of Maryland, College Park, MD 20740, USA; 22Novel Microdevices, Nucleic Acids, Baltimore, MD 21202, USA; michael@novelmicrodevices.com; 23Institute for Advanced Computer Studies, University of Maryland, College Park, MD 20740, USA; 24AgBiome, 104 TW Alexander, Research Triangle, NC 27709, USA; cpeperanney@agbiome.com; 25Bioscience Division, Bikini Atoll Road, Los Alamos National Laboratory, Los Alamos, NM 87545, USA; migun@lanl.gov; 26Center for Dark Energy Biosphere Investigations, University of Southern California, Los Angeles, CA 90089, USA; tully.bj@gmail.com; 27AAAS Science & Technology Policy Fellow, National Institutes of Health, Center for Information Technology, 6555 Rock Spring Drive, Bethesda, MD 20817, USA; valerie.virta@nih.gov; 28Department of Marine and Environmental Biology, University of Southern California, Los Angeles, CA 90089, USA; jakeweis@usc.edu; 29DNANexus, 1975 W El Camino Real #204, Mountain View, CA 94040, USA

**Keywords:** data federation, CRISPR, protein domain, metagenomics, virus, genome graphs, HIV-1

## Abstract

Viruses represent important test cases for data federation due to their genome size and the rapid increase in sequence data in publicly available databases. However, some consequences of previously decentralized (unfederated) data are lack of consensus or comparisons between feature annotations. Unifying or displaying alternative annotations should be a priority both for communities with robust entry representation and for nascent communities with burgeoning data sources. To this end, during this three-day continuation of the Virus Hunting Toolkit codeathon series (VHT-2), a new integrated and federated viral index was elaborated. This Federated Index of Viral Experiments (FIVE) integrates pre-existing and novel functional and taxonomy annotations and virus–host pairings. Variability in the context of viral genomic diversity is often overlooked in virus databases. As a proof-of-concept, FIVE was the first attempt to include viral genome variation for HIV, the most well-studied human pathogen, through viral genome diversity graphs. As per the publication of this manuscript, FIVE is the first implementation of a virus-specific federated index of such scope. FIVE is coded in BigQuery for optimal access of large quantities of data and is publicly accessible. Many projects of database or index federation fail to provide easier alternatives to access or query information. To this end, a Python API query system was developed to enhance the accessibility of FIVE.

## 1. Introduction

While the sharp reduction in the cost of sequencing over the past 15 years [1] is leading to the progressive democratization of biomolecule sequencing and experimental data production, the resulting data influx represents a nightmare of data storage, management, accessibility, and analysis. As of November 2019, the Sequence Read Archive (SRA) contained almost 13 petabases of open information, with close to 20 more petabases in the queue [2] (data growth is periodically updated at [3]). Tackling the complexity and depth of this information is daunting. Despite the existence of stable general repositories (Genbank, ENA, DDBJ), a growing number of specialized databases are leaving the results of biological experiments (mostly sequence data) disconnected, sparse, disorganized, and often inaccessible. Data accessibility and data federation, the virtualization of sparse databases into a common platform, represent the most important assets to the wider scientific community. There have been previous attempts to federate such databases and to make them open access, specifically for viral sequences [4]. 

The National Institutes of Health (NIH) continues to promote such efforts through the Science and Technology Research Infrastructure for Discovery, Experimentation, and Sustainability (STRIDES) Initiative, which “provides cost-effective access to industry-leading partners to help advance biomedical research” [5]. These partnerships enable access to rich datasets and advanced computational infrastructure, tools, and services. The STRIDES Initiative is one of many NIH-wide efforts to implement the NIH Strategic Plan for Data Science, which provides a roadmap for modernizing the NIH-funded biomedical data science ecosystem [6]. The National Center for Biotechnology Information (NCBI) [7] leverages their participation in the STRIDES Initiative in part by organizing and supporting a series of events, such as hackathons and codeathons, to engage researchers and general users of NIH resources to improve NIH resources [8]. These events usually span a three-day period and are geared towards addressing a specific research problem or topic. Through these events, the NIH receives active feedback from their user community on existing resources that helps improve the quality and output of future NCBI products. Typically, milestones or minor objectives are brainstormed during an online organizational meeting before the event. These objectives form the core of the working groups during the event. Over the course of the three days, each working group produces a solution to a specific task, often collaborating and integrating their solution with the products from the other working groups. The event concludes with working groups presenting their solutions.

NCBI also leveraged their participation in the STRIDES Initiative by moving SRA to the cloud. SRA is the largest source of open, publicly available next-generation sequencing (NGS) data from diverse biological sources. SRA serves as an umbrella for a variety of sequencing experiments (e.g., amplicons, whole genome sequencing, and environmental metagenomics) from different platforms (namely IonTorrent, Illumina, Oxford Nanopore, and PacBio) and applications. The first collaborative attempt to annotate and index SRA datasets in bulk was conducted as part of the inaugural of this series of events, the Virus Hunting Toolkit (VHT) [4]. This first event (VHT-1) challenged users to harness the power of the Google Cloud environment to test and develop bioinformatics pipelines to identify all viruses, including previously characterized and novel, in existing publicly available SRA datasets. The working groups were organized by specific tasks to emphasize the exhaustive separation of known viral diversity from the bulk data, and the identification of possible new viral sequences: data selection, taxonomic and cluster identification, and annotation of domains and genes. From that first codeathon, a set of 5.5 × 10^7^ reassembled contigs, from 2953 SRA entries, were produced (see Table 1 on accessing these resources) [4]. All contigs were classified and annotated, and this information was integrated into a complete dataset [9]. Despite the efforts made, some areas were not covered in the final annotation files, such as virus–host pairing predictions. An additional impending limitation was developing concise strategies to store and visualize sequencing data from related biological entities. In our second series of Virus Hunting in the Cloud 2.0 (VHT-2), we present here the first Federated Index of Viral (Sequencing) Experiments (FIVE). 

In contrast to traditional sequence data databases, federated indices do not store raw sequence data. Instead, federated indices store the results from different sequence analyses. For example, to identify putative novel viruses in SRA contigs, a federated index would store and link results from several analyses on these contigs, allowing researchers to quickly identify SRA contigs of interest without the need to perform the same extensive underlying analyses. Therefore, the output of FIVE is a collation of analysis results from different methods indicating if and where a sequence contains viral or virus-like signals, and not a single similarity score to a known virus sequence or structure. 

This novel resource indexes sequences, metadata, and hyperdata from the VHT-1 hackathon, sparse databases, and online resources. More than 2953 SRA entries assembled during VHT-1 [4] were reanalyzed and their annotations included in FIVE. Additionally, existing databases (i.e., taxonomy, CRISPR, etc.) were federated into our index, expanded with novel annotations. Finally, FIVE includes the first attempt to condense viral genome variation (nucleotide diversity along the genome) into an indexable genome graph [10], using HIV-1 as a proof-of-concept. The emerging field of genome graphs can provide an efficient method to summarize and index sequence diversity data for a single species [11]. Unlike previous efforts, FIVE is a publicly accessible index where several methods were built to easily query and retrieve information from the index. The index accession methods were wrapped into an easy-to-use Application Programming Interface (API) written in Python to further improve its accessibility. The FIVE index links SRA with accurately assembled contigs, viral and host taxonomy annotation, protein/functional annotation (assisting taxonomy identification), and virus–host pairing predictions. FIVE was generated by (i) federating or mining existing datasets, (ii) implementing novel methods for annotation, and (iii) indexing and improving data access. Several teams were formed, focusing on different aspects to generate FIVE. Resulting from the VHT-2, four core aspects of FIVE can be distinguished: (i) protein domain recognition and computation scalability, (ii) virus–host pairing prediction, (iii) viral genome variability indexing in genome graphs, and (iv) index structure and accessibility. 

New functional and taxonomic annotation methods were developed, covering similarity search by alignment, probabilistic models (hidden Markov models [HMMs]), and *k*-mer distances. These methods were added to the extensive information in our previously published SRA annotations [4]. Known virus–host pairings were federated from existing databases for both eukaryotic and prokaryotic viruses and expanded through novel phage annotation and CRISPR profiling. Besides taxonomy or function, genome variability is another desirable layer of information to consider for virus diversity. Mutations in viral genomes can help to trace geographical patterns, distinguish closely related viruses, or to predict protein functionality and therefore infective properties. Genome graphs are an emerging tool to compress genomic information (e.g., variants) from closely related genomes into more compact formats compared to multiple linear reference genomes or multiple sequence alignments. Despite their compactness, if many variants exist, such graphs can become considerably large. Such diversity is of interest to have indexed together with other types of data. With FIVE we present a proof-of-concept of viral genome graphs indexing and systematization. We refined a new and compact approximate *k*-mer graph creation tool, SWIft Genomes in a Graph (SWIGG), which was used to model the genome variation of full-length HIV-1 reference genomes. Finally, FIVE is intended to be a free, public, usable database by a broader audience, therefore besides the publication of FIVE as a public BigQuery index, we designed an easy-to-use Python application programming interface (API), and some methods, to query FIVE. This index is the first attempt to centralize and federate viral (meta)data and annotations directly from SRA. As such, FIVE is distinct from, yet complementary to, other resources such as ViPR [12] or NCBI Viral Resources.

## 2. Materials and Methods 

### 2.1. Protein Domain Recognition and Computation Scalability

To improve the contig annotation index in FIVE, we evaluated 2953 datasets from VHT-1 [4,13] against 2082 viral-specific protein domains selected from the CDD database (see [14]). We designed two pipelines: (i) Reverse Position Specific tBLASTn (RPS-tBLASTn) [15] and (ii) Mash pipelines [16] to identify known viral protein domains within VHT-1-assembled contigs (see Table 1). RPS-tBLASTn is a robust method for domain detection, although its computational demands render it challenging for large-scale data annotation, such as in VHT-2. In this context, a distance estimator, based on protein sequence sketches, can be used to identify protein domains with less computational demand. Mash [16], a metagenome distance estimation tool (based on MinHash dimensionality reduction), was used with amino acid *k*-mer size = 6. A subset of 728 datasets was used to compare RPS-tBLASTN and Mash [17] to assess the performance of the distance estimation. Recall percentages for the Mash pipeline were calculated per dataset by dividing the ‘true positive’ viral CDDs by the sum of ‘true positive’ and ‘false negative’ viral CDDs. Precision percentages for the Mash pipeline were calculated per dataset by dividing the ‘true positive’ viral CDDs by the sum of ‘true positive’ and ‘false positive’ viral CDDs. A CDD hit was considered either a true positive if it was retrieved in the same dataset, a false positive if it was retrieved by the Mash pipeline but not by the RPS-tBLASTn pipeline, or a false negative if it was retrieved by the RPS-tBLASTn pipeline but not by the Mash pipeline. Clustering was performed on the Canberra distance matrices derived from the domain counts matrices using base R function *hclust* (stats package v3.5.3) [18]. Correlation between both matrices was calculated with the Mantel test implemented in the *ade4* R package (v1.7.-15) [19]. Normalized Robinson–Foulds metrics were calculated with the *RF.dist* function in the *Phangorn* R package (v2.5.5) [20], and entanglement values and tanglegrams were calculated and plotted using the *dendextend* package (v1.13.4) in R (v3.6.2) [21]. Additionally, HMMER (v3.1) [22] was explored as a short read taxonomic annotator, providing an alternative to a computationally intensive de novo assembly step.

A schematic of both pipelines can be seen in Figure 1, each of which relies on two sets of inputs: (i) a set of 2953 datasets containing assembled contigs constructed during VHT-1 [4], and (ii) a selected set of 2082 virus-associated Conserved Domains Database (CDD) entries (personal communication from J. Rodney Brister, NCBI RefSeq [23,24] Appendix A).

We matched CDD entries against the dataset using RPS-BLAST, with 6-frame translation (RPS-tBLASTn) [25]. We filtered results to include only higher-quality output hits with e-value ≤ 1 × 10^−10^ and coverage threshold length ≥ 50 nucleotides.

In addition to RPS-BLAST and Mash, we tested the feasibility of domain detection using HMMER [22] directly against short sequence reads. We generated a simulated dataset of 1000 reads (each 150 bases long) by randomly extracting simulated reads from a complete genome of Human Herpesvirus 1 (HHV-1, GenBank accession JN555585). The DNA sequences were translated into the six possible reading frames using Biopython [26], yielding 6000 short “peptides” (~50 amino acids). This simulated dataset enabled us to evaluate HMMER (*hmmscan*) for searching for domains in short reads.

### 2.2. Virus–Host Pairing Prediction

FIVE included an index for virus–host pairing (which viruses can infect which organisms). Initially, we federated a reference dataset of experimentally confirmed viral–host pairings, generating 22,896 known virus–host pairings from the NCBI Virus Variation Resource database [27] and supplementing this with records from PhagesDB [28]. Missing information from PhagesDB, including host and virus taxonomic identification numbers and taxonomic lineages, was retrieved using the NCBI Taxonomy Browser [29] and incorporated into the index (see custom scripts in [30]). In several instances, phage genomes without a specified host in the database possessed a bacterial genome in the phage name that allowed for relationship inference.

The federated index was expanded upon using CRISPR spacer connections. A large-scale CRISPR spacer database was generated from the initial federated index and four datasets generated using distinct sources: The first dataset of CRISPR spacers was CRISPRCasdb ([31]; accessed 6 November 2019), where spacers were identified using the tool CRISPRCasFinder [32] from “reference” and “representative” microbial genomes available in RefSeq [23]. The second dataset of CRISPR spacers [33] was identified from all prokaryotic assemblies in RefSeq [23] (December 2017) using CRISPRDetect [34]. The third dataset of CRISPR spacers was identified from 24,345 metagenome-assembled genomes (MAGs) of the human microbiome [35] using MinCED [36], based on the CRISPR Recognition Tool [37] (parameters: -spacers -gffFull). The fourth dataset of CRISPR spacers was identified from the 24,706 species-representative sequences in Genome Taxonomy Database (GTDB) [38] using MinCED. All genomes/MAGs were provided using standardized taxonomy, based on the GTDB taxonomy. The resulting CRISPR spacer database was searched against the initially federated index of known virus–host pairings using BLASTn, with parameters set to account for the short size of the spacer regions (parameters: -task blastn-short, -evalue 0.01, -outfmt 6, -gapopen 10, -gapextend 2, -penalty “−1”, -word_size 7, -dust no).

### 2.3. Viral Genome Diversity Indexing in Genome Graphs

As introduced earlier, we used genome graphs as proof-of-concept to index viral genome variability for a given set of closely related viruses. We used SWIft Genomes in a Graph (SWIGG) (commit 48c4661), a nascent genome graph builder, as a backbone for FIVE’s own implementation [39]. In short, SWIGG creates genome graphs using *k*-mers from input genome sequences. The *k*-mer length can be set by the user, and *k*-mers used for genome graphs can be excluded by fine tuning the maximum and/or minimum *k*-mer counts within or across analyzed sequences. We added the functionality to parse metadata from sequence headers and incorporate both into the individual nodes of the genome graph. We used Human Immunodeficiency Virus 1 (HIV-1) as a test case for the integration of viral genome diversity into FIVE (sequences available at [40]). HIV-1 was selected because at the time of submission, it was the most well-studied human pathogen, having the most high-quality full- or near full-length genomes available [41], with robust feature annotations including structural features at the proviral DNA, viral RNA, and viral protein levels.

To balance sequence diversity with representative HIV-1 sequences, we used 170 HIV-1 reference genome sequences from the Los Alamos National Laboratory’s HIV Sequence Database ([42]; accessed on 4 November 2019). To retrieve these sequences, curated alignments were accessed with the following parameters: from “Alignments”, “Curated alignments” was selected; Alignment type = “Subtype reference”; Pre-defined region of the genome = “GENOME”; subtype = “ALL”; DNA/protein = “DNA”; year = “2010”. This number was narrowed down from 170 to 167 after removing “cpz” or SIV sequences. We also used a subset of these yielding 39 sequences by changing subtype = “M group without recombinants (A–K)”. The implementations require Python v3.6 or higher and the Python package *NetworkX* (v2.4) [43]. 

### 2.4. Index Structure and Accessibility

We indexed data generated by working groups during the second codeathon (“Virus Hunting in the Cloud 2.0”) in a relational database format on Google Cloud’s BigQuery, called Federated Index of Viral Experiments (FIVE). This index can be visualized as distinct silos of the data generated by each of the analysis pipelines presented in this manuscript. Data was parsed [44], loaded into BigQuery using google-cloud-sdk (v288.0.0) tools, and SQL-like manipulation and subsetting of the tables was performed (scripts available at [45]). Data loaded into FIVE is made accessible to the public (see Table 1).

One of the most important aspects of an index is accessibility. BigQuery is a relatively new framework and hence it was decided to generate a series of queryable actions, answering the most frequent research questions. To facilitate this research, a Python-based API was developed. This API, called viral-index (v0.0.3) (Figure 2), is freely available to download from PyPI [46]. Installation of the viral-index module requires Python 3.7 and the Python packages pip (v20.0.2) and virtualenv (v20.0.18). The viral-index module relies on google-cloud-bigquery (v1.27), google-auth (v1.21.1), and twine (v3.2.0) Python3 modules, which should be downloaded as part of the viral-index module dependencies. After installation, the user must add the path to their google credentials, as system variable “GOOGLE_APPLICATION_CREDENTIALS”. These credentials allow the user to access the BigQuery databases (detailed instructions at [47]) and query the federated indexes using a range of functionalities implemented in the viral-index module, which we describe in this manuscript. It is important to note that the viral-index API module supports data retrieval but not data manipulation. The viral-index module returns the data as standard list() or dict() objects that can be easily manipulated in order to carry out further analysis. Additionally, this framework enables incorporation of new or updated datasets when they become available.

### 2.5. Data and Software Availability

A broad and detailed explanation of each method section can be found in the corresponding GitHub projects (Table 1). Each project contains complete instructions to fully reproduce the data generated and to reproduce FIVE. Links to the VHT contigs and FIVE are also made available (Table 1).

## 3. Results and Discussion

### 3.1. Protein Domain Recognition and Computation Scalability 

We ran the RPS-BLAST pipeline on contigs derived from 2953 public sequencing datasets assembled in VHT-1 [4] (see Table 1 for access to the contig list and contig repository from Google Cloud), but we found it to be too computationally expensive to run with the complete CDD database over three days. Instead, we used a subset of 2082 viral CDD models. 

Since RPS-tBLASTn does not allow for query parallelization natively, we tried several parallelization strategies for RPS-tBLASTn in order to scale up its performance. Subsequently, we attempted to parallelize the RPS-tBLASTn search using a procedure whereby we divided the database into 60 segments and later combined the results for each segment. Splitting the database and later rejoining results affected the search space, and therefore the e-value of our results. While appropriate for testing purposes, for production runs we strongly recommended to use the *dbsize* parameter to account for the changed search space as a correction factor.

Among the 55,503,968 contigs that we searched against the viral specific CDDs [24], 10% of the contigs (5,606,754 from 2745 SRA datasets) had at least one CDD hit with e-value ≤ 1 × 10^−3^. Using a more stringent e-value ≤ 1 × 10^−10^, the number of contigs having at least one viral CDD was reduced to 0.5% (278,725; from 2534 entries). Hit distribution varied enormously; some contigs had multiple CDD hits, peaking at 22,560 hits (Contig ID = NC_003663.2:1.224499, Cowpox virus), but the majority of contigs (77.3%) had one unique CDD hit. The most common CDD was CDD:222853 (a transposase specific to the *Caudovirales* lineage).

In parallel, we tested the (meta)genomic distance estimation tool Mash (MinHash dimensionality reduction) [16,49] on predicted proteins (obtained through Prodigal protein prediction) from the contigs derived from 728 datasets [17], a subset of the 2953 datasets used in the previous RPS-tBLASTn analysis [50]. Mash’s default amino acid *k*-mer length of *k* = 21 (for both of its input sketches) retrieved almost no hits, so a length of *k* = 6 was chosen arbitrarily. We could not assess other *k* values (which may have yielded better estimations) due to time restrictions.

A representative subset of 728 datasets was used to compare the Mash and the RPS-BLAST pipelines. Out of 728 datasets, the Mash pipeline had an order of magnitude fewer hits (133,452 hits) than with RPS-tBLASTn (2,574,452 hits). The Mash pipeline was found to be substantially less sensitive than the RPS-tBLASTn analysis, with an average recall of 15.3% and a precision of 37.0%. Despite the low recall value, Canberra distance matrices retrieved from the Mash datasets and RPS-tBLASTn datasets were strongly correlated (Mantel test, p-value = 0.0009). Additionally, hierarchical clustering implied that, despite the loss of global structure in the dataset (as shown by a Robinson–Foulds distance of 0.91 and an entanglement of 0.2), the Mash pipeline can be used as a fast tool to quickly identify datasets containing roughly similar viral domains (see Figure 3). While the Mash pipeline performed significantly faster than RPS-tBLASTn, the Prodigal translation step still represented a significant bottleneck for scaling. Faster translation algorithms would increase the feasibility of the Mash pipeline. While the fast Mash pipeline method proved to be promising as an alternative to more computationally heavy methods, such as RPS-tBLASTn, a lack of recall and precision deemed this method unsuitable for exhaustive research.

The HMMER pipeline (hmmscan against Pfam-A v33.0) was applied to a simulated viral dataset of short reads. A total of 75 herpesvirus-specific domains were detected, with e-values ≤ 1 × 10^−3^. On a node with 64 2.30 GHz cores, a total of 136 peptide sequences per minute were scanned for domain detection. In order to improve comparability, the same procedure was applied to search domains in a real, non-simulated sequence dataset (Illumina paired-end data; SRA: ERR1137115). A total of 1000 randomly selected reads were extracted, translated into the six possible frames, and scanned for domains using the HMMER pipeline. For this dataset, 125 peptides were searched per minute, detecting 109 domains. Despite its ability to detect viral domains in short sequences, the HMMER pipeline performance did not scale well enough for datasets containing millions of reads over the course of three codeathon days. In order to deploy the HMMER pipeline for such large datasets, we propose the following: (i) collapsing identical or near-identical sequences to reduce redundancy; (ii) splitting translated frames yielding truncated peptides into distinct peptides, using stop codons as peptide boundaries; and, (iii) filtering amino acid sequences by length (≥50 amino acids) to decide whether to accept them as queries for domain detection. Applying these premises might still not be enough to yield reliable resources over the course of an event like a codeathon but might be sufficient to yield results under reasonable research time (e.g., over a week).

### 3.2. Virus–Host Pairing Prediction

To establish a baseline for known virus–host pairings, we federated several resources to act as references for experimentally confirmed and inferred interactions for the FIVE. In addition, we included queryable datasets designed to detect putative viral elements and linkable to a putative host. These putative pairings included phages and prophages mainly, which may help to understand prophage variability in well-categorized host systems. We federated existing databases providing information on hosts (including Bacteria, Archaea, and Eukarya) and the identity of confirmed viral pairings from PhagesDB and the NCBI Virus Variation Resource database. The aforementioned resources were combined, expanded, and standardized to produce a comprehensive virus–host pairing index, containing 44,975 virus–host pairs (29,847 unique viruses and 7974 unique hosts) that can be queried from FIVE.

Identified CRISPR spacers from four datasets (CRISPRCasdb, RefSeq, 24,345 human microbiome MAGs and 24,706 GTDB species-representative sequences) were curated and compiled into a comprehensive CRISPR spacer database, with 1 million unique spacer sequences linked to a formalized host taxonomy. CRISPR spacers were compared against the 29,847 unique viruses with known hosts identified in the virus–host pairing index. In addition, the CRISPR spacers were compared against 2953 raw datasets selected from NCBI’s SRA used in VHT-1 [4].

### 3.3. Viral Genome Diversity Indexing in Genome Graphs

To demonstrate the ability of FIVE to index a wide variety of data types, we made genome graphs with HIV-1 reference genomes. Genome graphs facilitate the analysis of the diversity of a set of closely related sequences by counting and connecting *k*-mers from multiple sources like full viral genomes or virus segments. Examples of our HIV-1 genome graphs are shown in Figure 4. The FASTA header for each sequence analyzed to create the graph was extended using brackets as key value pairs, i.e., “>Accession [key=value] [key=value]”. Specifically, we indexed the viral diversity using the metadata of individual *k*-mer graph nodes. This metadata was later included in the individual graph nodes (not shown). The sequences were assembled into a graph using the Python3 package *NetworkX* and stored as a GraphML file that can be visualized and further analyzed using free open-source software such as Gephi [51] or Cytoscape [52,53].

### 3.4. Index Structure and Accessibility

Contig annotations and graph data derived from protein domain recognition, viral-specific HMM, virus–host pairing, and HIV-1 genome variability were produced in tabular format (as detailed at the end of each previous sections). Each table was loaded into the FIVE BigQuery Google Cloud index [45] to be queryable. FIVE consists of seven interconnected tables (accession2species [3,174,289 entries], combined_known_interactions [46,979 entries], cdd_data [2,765,472 entries], spacer_db [18,521,874 entries], domains_viral_cds_tblastn [26,902,443 entries], and hiv_a_jrefs_k41_t2 [247 entries]), as seen in Figure 5, and can be freely accessed at [54] (link provided in Table 1).

A range of different functions are implemented in the viral-index API module that enable easy access to FIVE. The viral-index can search the federated indexes by SRA run ID, virus and host taxonomy ID, and CRISPR spacer sequences (as seen in Table 2).

Virus–host pairs can be searched using the *get_host_for_virus_taxonomy* function that takes a NCBI virus taxonomy ID as input and returns all hosts that the given virus could infect. In order to perform the inverse search, *get_viruses_for_host_taxonomy* can be applied, and it may allow users to search for viruses that could infect a given host taxonomy ID; this search can be expanded to incorporate the protein domain-based information. The *get_potential_hosts_for_virus_domain* function integrates the data generated for the protein domains and the virus–host interactions. Thus, it allows searching potential virus hosts that viruses with a specific domain could infect by searching the federated data using a CDD domain ID. Other domain-based functions include (i) *get_SRAs_where_CDD_is_found* and (ii) *get_domains*, whereby users can retrieve (i) specific SRA studies where a virus-specific domain is present and (ii) the viruses that may contain a domain, respectively. The former function can be used to get a snapshot of virus domains in several SRA studies analyzed in VHT-1 [4]. Two additional functions are implemented to retrieve CRISPR signature-based spacer indexes. The *get_spacer_seqs* function enables the users to fetch all spacer sequences present in the spacer datasets for a given taxonomy ID. The *get_metadata_from_spacer_seq* function retrieves spacer ID, spacer sequence, GenBank accession, and taxonomy identification of organisms where the given spacer sequence is present. An example, showing how to use the *viral-index* to retrieve all viruses that infect pigs, is provided at [55].

It is important to note that data integration is becoming the norm; powerful analyses can be performed when it is possible to interlink data generated and enriched with multiple layers of known and novel information. The *viral-index* API enables researchers to interrogate increasingly sophisticated biological questions from FIVE through the multi-layer information available in this federated database indexes.

## 4. Conclusions and Future Directions

During this three-day continuation of the VHT codeathon series (VHT-2), a new integrated and federated viral index was elaborated. This Federated Index of Viral Experiments—FIVE—integrated new functional and taxonomy annotations, novel virus–host pairings, and for the first time, introduced virus genome diversity as genome graphs. Additionally, FIVE contains a federation of annotations and pairings from pre-existing sources. As per the publication of this manuscript, FIVE is the first implementation of a virus-specific federated index of such scope.

Several metagenomic annotation pipelines were developed and tested, building on top of the foundations laid out in previous editions. Three pipelines for annotation of viral contigs through protein domains were proposed: (i) RPS-tBLASTn, (ii) Mash, and (iii) HMMER. Results showed differences in recall, accuracy, and speed between RPS-tBLASTn and Mash. RPS-tBLASTn may have been more computationally expensive than Mash, but it had better recall and overall accuracy. Additionally, as evidenced from VHT-1 [4], HMMER searches could not be fully scaled in a cloud environment, representing a bottleneck in protein domain classification using HMMs. Despite the ability of RPS-tBLASTn to be pseudo-parallelized, the main bottleneck for high-throughput cloud computing was scalability. Based on the current results, the RPS-tBLASTn pipeline was the best-performing implementation out of the three and the one we recommend for other large-scale cloud computing initiatives.

We made an additional effort to expand and federate not only the annotation tools for viral datasets, but also its taxonomical pairing with a given host. This original work expanded the number of known viral–host taxonomical pairings by 129% over VirHostNet 2.0 (release 1/2019) [56], by integrating a federated high-confidence dataset and a novel dataset based on de novo assignations. The high-confidence dataset is based on a federation of the NCBI Virus Variation Resource and PhagesDB databases. The novel dataset is built with predicted past pairings using CRISPR spacers. An expanded CRISPR dataset was created with 1M unique spacers to identify previously unknown relationships between complete viral genomes with taxonomy and the CRISPR spacer isolation source.

One of the last challenges during the codeathon was to start developing a pipeline and indexing strategy for virus genome diversity. It is known that genome graphs can be used to efficiently summarize known virus genome diversity; thus, as a proof-of-concept, we decided to build an HIV-1 genome diversity graph to index the variability into a federated index, such as FIVE. The two main challenges were (i) finding appropriate *k*-mer settings and (ii) adding multiple metadata values to virus genome diversity graphs. Proper attachment of metadata is crucial for indexing datasets and ensuring that data is findable, accessible, interoperable, and reusable (FAIR) [57]. Metadata can also be overlaid onto genome diversity graphs to improve functional interpretation. Metadata was added from the analyzed sequences but was inadequate when evaluating features within graphs. A limitation of available reference genomes included nonuniform feature annotation formatting. Follow-up work will be needed to analyze the influence of SWIGG parameters on subsets of HIV-1. Linking individual *k*-mer nodes to component sequence annotations will further enhance the possibility to mine the structural information represented in graphs and to connect it to biological function. In the current state, creating genome diversity graphs is a convoluted process involving iterative testing of multiple parameters and visual inspection of their resulting graphs. Visualizing multiple metadata layers simultaneously becomes challenging and, ultimately, the manual analysis of multiple and complex graphs becomes an unfeasible task. It will be important to develop automated assessments of virus diversity graphs to adjust construction parameters, develop visualization methods for multiple metadata values, and create methods to automate graph analysis. Input from viral genomics is needed in order to standardize a genome diversity graph format.

Annotations and metadata from the different projects were integrated into the FIVE BigQuery index and later made queryable, making it the first implementation of a virus-specific federated index that is easily accessible and queryable. As per the development of a Python-based API, the FIVE can be *de facto* used by a larger part of the research community (possessing basic scripting abilities). Accessibility and ease of implementation are often the limiting factors for the broad use of public resources. A graphical user interface (GUI) is under discussion to further broaden accessibility. The final aim is to link FIVE to other widely used viral and host resources, such as those supported NCBI, centralizing the resource and improving its connectivity to other services. Efforts are underway to maintain FIVE through annual updates and continuous federation.

## Figures and Tables

**Figure 1 viruses-12-01424-f001:**
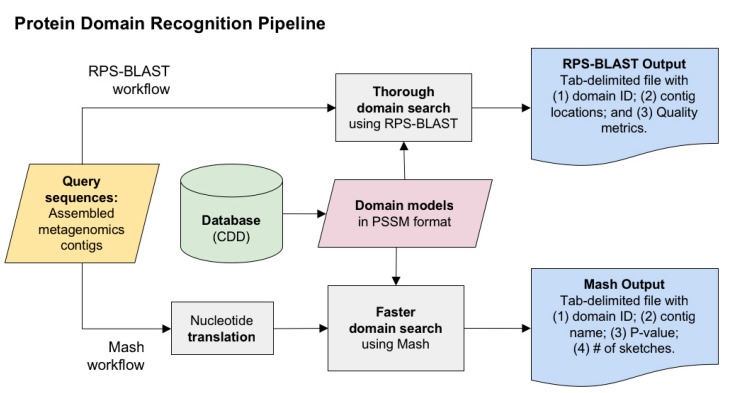
Protein Domain Recognition Pipeline. Using 2082 entries from CDD (Conserved Domains Database) domain models in PSSM (Position-Specific Scoring Matrix) format, we tested two pipelines: RPS-BLAST and Mash. RPS-BLAST, with known domain models matched against assembled contigs, is accurate but computationally expensive. The Mash pipeline, which is significantly faster and can be applied directly on unassembled reads, was also tested.

**Figure 2 viruses-12-01424-f002:**
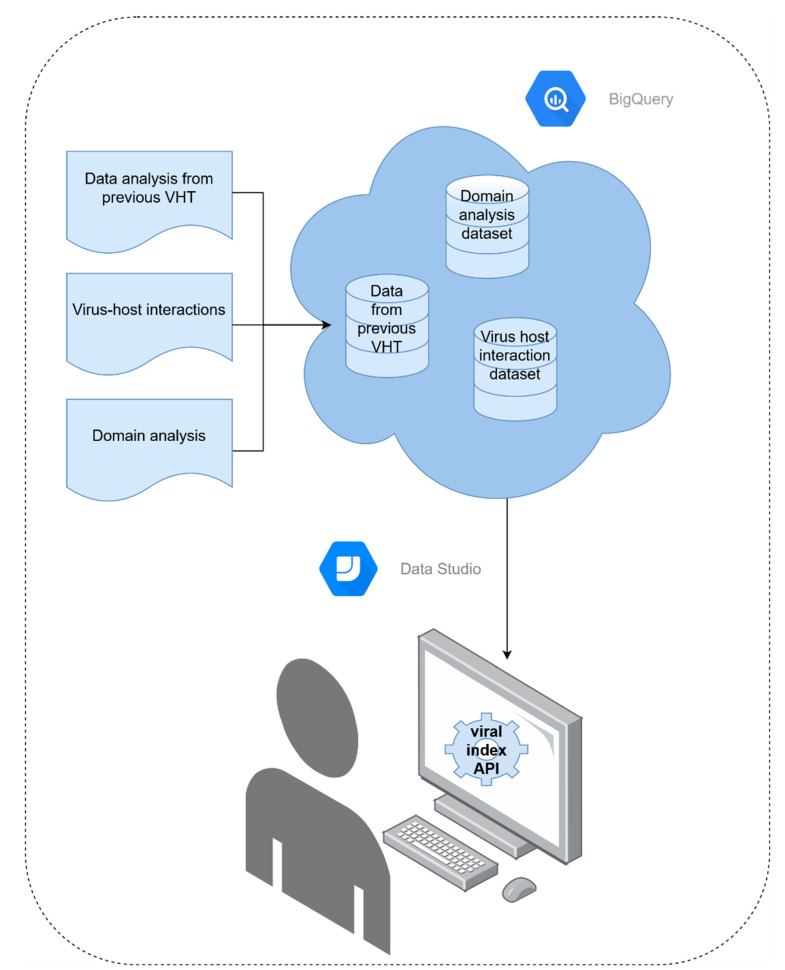
A schematic representation of Federated Index of Viral Experiments (FIVE) implementation, and interactions with users, enabled through the viral-index Application Programming Interface (API). Viral information generated in both codeathons is indexed in BigQuery on FIVE, accessible from Google Cloud, which can be easily queried using the viral-index API [48]. This API enables users to perform a range of flexible searches on the FIVE databases with minimum code.

**Figure 3 viruses-12-01424-f003:**
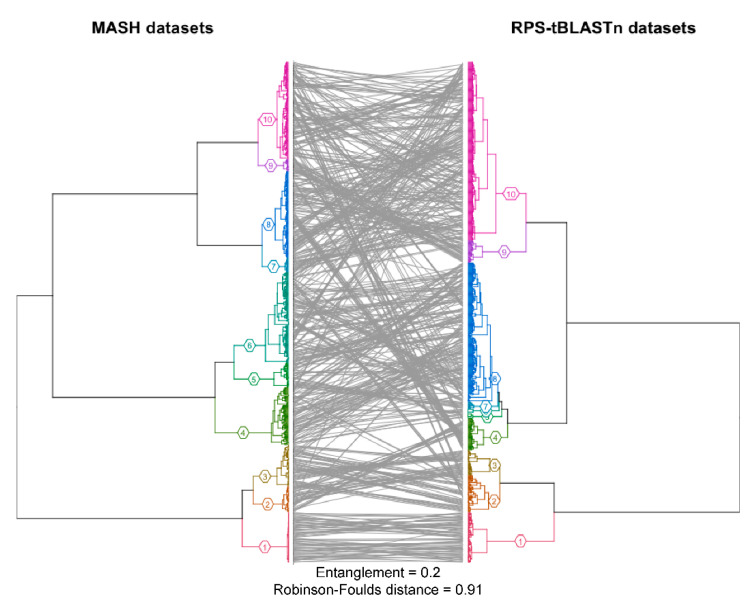
Tanglegram depicting hierarchical clustering performed on the Canberra distance matrices derived from the domain counts matrices of both Mash and RPS-tBLASTn pipelines. Both dendrograms are colored by their cluster id with *k* = 10. Base R function hclust was used to generate the clustering [18]. Correlation between both matrices was calculated with the Mantel test implemented in the ade4 R package [19]. The entanglement value and plot were generated with the Entanglement and Tanglegram functions implemented in the dendextend package [21]. Robinson–Foulds distance was calculated using the RF.dist function implemented in the Phangorn package [20].

**Figure 4 viruses-12-01424-f004:**
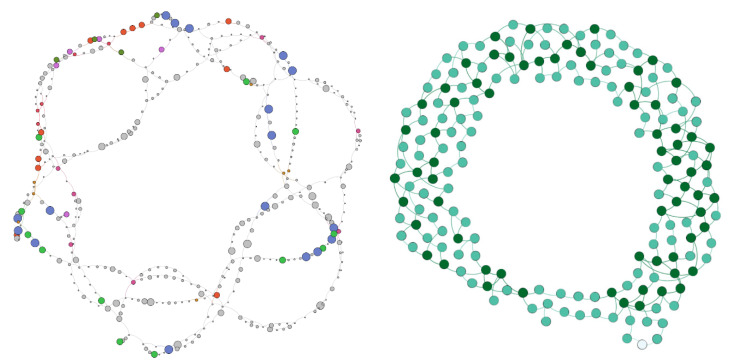
(Left) HIV-1 reference genome graphs generated with SWIft Genomes in a Graph (SWIGG) with annotated *k*-mers/nodes. Number of input sequences (*n*) = 167. Node color corresponds to taxonomic distribution of *k*-mer. Size of nodes is proportional to occurrence of taxonomic category. (Right) HIV-1 subtypes A–J (*n* = 39), *k*-mer size = 41, threshold ≥ 2. Note that both example graphs are circular, which may represent the fact that common nodes occur within long terminal repeats (LTRs). Most of the HIV references used in this work were modeled after the proviral sequence, which includes 5′ and 3′ LTRs.

**Figure 5 viruses-12-01424-f005:**
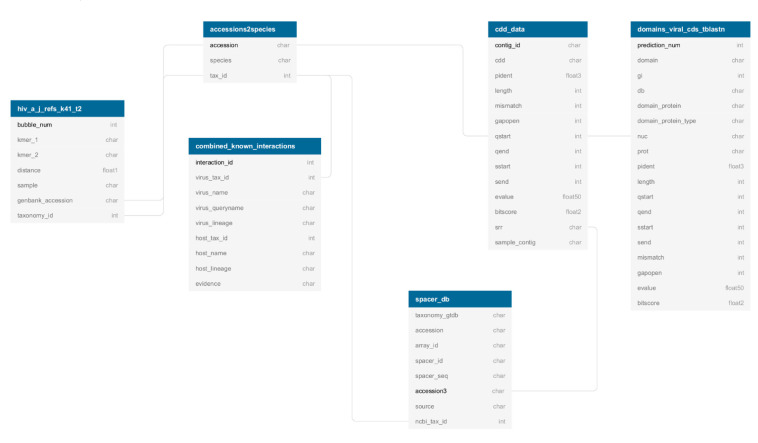
FIVE index schema. Each table (boxes) represents the output from the different annotation efforts towards FIVE. For each table, the title of the table is white in a blue rectangle (*accession2species*, *combined_known_interactions*, *cdd_data*, *spacer_db*, *domains_viral_cds_tblastn,* and *hiv_a_jrefs_k41_t2*), immediately followed by the field names or categories for that given table. Each line corresponds to a field, in which the first column gives the abbreviation name for the content of the field and the second column the format of the content (int for integers, char for strings of characters, float and decimals). Primary keys for each table are found in bold. It is possible to both access each one of the tables independently and to link primary keys from one table to fields from another table, generating a link (in grey).

**Table 1 viruses-12-01424-t001:** List of repositories used in the generation of FIVE and its accessory information (contigs and FIVE link) hosted in GitHub and the first release of each repository frozen in ZENODO. VHT—Virus Hunting Toolkit.

Relevant Repositories	GitHub Project	Dataset Citation
Connor et al., 2019 VHT [4]	https://github.com/NCBI-Hackathons/VirusDiscoveryProject	10.3390/genes10090714
VHT contig list	https://github.com/NCBI-Hackathons/VirusDiscoveryProject/blob/master/contigs_readme.md	10.17605/osf.io/g9w8r
VHT contig repository	https://storage.googleapis.com/experimental-sra-metagenome-contigs	10.17605/osf.io/g9w8r
Protein domain recognition and computation scalability	https://github.com/NCBI-Codeathons/Domain_HMM_Boundaries	10.5281/zenodo.4027168
Virus–Host pairing prediction	https://github.com/NCBI-Codeathons/Host_Phage_Interactions	10.5281/zenodo.4027172
Viral genome diversity indexing in genome graphs	https://github.com/NCBI-Codeathons/Virus_Graphs	10.5281/zenodo.4027629
Index structure and accessibility	https://github.com/NCBI-Codeathons/The_Virus_Index	10.5281/zenodo.4027617
FIVE	https://console.cloud.google.com/bigquery?p=virus-hunting-2-codeathon&d=viasq&page=dataset	-

**Table 2 viruses-12-01424-t002:** Summary and description of primary viral-index API query functions.

Function	Description
get_viruses_for_host_taxonomy	Retrieve host(s) for a given virus taxonomy ID
get_host_from_virus_taxonomy	Retrieve virus(es) that can infect a given host
get_potential_hosts_for_virus_domain	Get all potential host(s) given a domain that is found in viruses
get_virus_host_interactions_from_confidence_level	Get all virus–host interactions for specified confidence level
get_SRAs_where_CDD_is_found	Get Sequence Read Archive (SRA) accessions of studies wherein a viral protein domain is found
get_domains	Find all domains present in a virus

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
