# Peer review of "NCBI’s Virus Discovery Codeathon: Building “FIVE” —The Federated Index of Viral Experiments API Index"

_viruses, 2020, doi:10.3390/v12121424_

Round 1
Reviewer 1 Report
I tried to understand the logic of the authors - but it was not easy
To give examples-
Section with Figure 1 - ... "recognition of protein domains"
If the recognition is done to classify metagenomic contigs as viral -
why to compare RPS-BLAST with MASH -- which was demonstrated to have poor prediction accuracy -
Better to compare RPS BLAST performance with the tools that are specialized on the virus sequence recognition - there are several ... VirFind, VirSorter, etc.
RPS BLAST performance (run time) would be much better if instead of 6-frame translation the genes in the contig were predicted first and then RPS BLAST would be used instead of RPS tBLASTn
Fig. 2 is missing
Fig. 3 is low informative - with information repeating itself twice
Given poor potential on MASH to solve the tasks - Figure 4 is becoming of very low interest
Fig. 5 meaning is not possible to understand
for instance --the legend says -"Node color corresponds to taxonomic distribution of k-mer." no single color is explained.
Fig. 6 would better belong to Suppl Materials (hard to understand without many additional clarifications of the field names)
Reviewer 2 Report
fine for me
This manuscript is a resubmission of an earlier submission. The following is a list of the peer review reports and author responses from that submission.
Round 1
Reviewer 1 Report
I do not see that the authors have reached the goal - creating a virtual (federated) database of viral sequences with uniform functional and taxonomy annotation
There is no statistics characterizing the structure, taxonomy and the numbers of entries in the database
I also did not find a clear and logical description of the pipelines that were supposed to be in place to process a large number - of order of 10 mln of contigs reassembled from 2,953 SRA entries
Several important sections are poorly written
Particularly section related to domain search pipeline (Lines 201-213)
The sequence of steps in this section is not well justified. The pipeline, as implemented, is split into two independent methods (Fig 1), one of them (Mash based) was eventually found to be substantially less sensitive than the second method (RPS-tBLASTn based) … an average recall of only 15.3%. – makes the second method not usable in practice unless the precision value is close to 100%. However, this second important measure – precision- was not mentioned at all.
The section does not make precise statements, for instance “Recall percentages were calculated per dataset by dividing the unique viral CDD’s identified by RPS-tBLASTn by the unique viral CDD’s identified by the Mash pipeline.” (lines 205-207)
Line 250 --- construction of whole genome HMMs, not well defined concept and method of design (needs multiple alignments of viral genomes) this has to be explained before coming up with a database of whole genome HMMs … a construct possible may be for short genome viruses but may not be so useful model for viruses with long genomes
Lines 265-271 Another critical section related to Taxonomy-domain-integration pipeline was not well written; it is impossible to reconstruct the logic and to see justification for the steps taken
Only fragments of the database were built – like virus graphs, which concept was not clearly explained, particularly its usefulness
Lines 530-535 – was the database built? – There is no link to a full size database, which seems to be a future work
Reviewer 2 Report
In this article Carreras et al. depict the works performed during the Virus Hunting Toolkit codeathon during which FIVE a federated viral database was elaborated. The initiative is particularly outstanding and need to be promoted. The article is well written and describes what has been done with, via a github repository, the open access to the methods developed.
The reviewer consider this article must be taken has it is, i.e. the description of a work that has been done and can not be done again.
As a virologist some aspects of what has been attempted still remain totally obscure to me (section 3.4 virus graph fig 5) but a detailed explanation is probably out of the scope of this article. May be citing some references would help virologist to understand the rational of this virus graph working group.
minor comments:
P14 L 513 searching potential instead of searaching potential
P15 L556 cloud computing instead of could computing